# A Novel Hand Teleoperation Method with Force and Vibrotactile Feedback Based on Dynamic Compliant Primitives Controller

**DOI:** 10.3390/biomimetics10040194

**Published:** 2025-03-21

**Authors:** Peixuan Hu, Xiao Huang, Yunlai Wang, Hui Li, Zhihong Jiang

**Affiliations:** National Key Laboratory of Autonomous Intelligent Unmanned Systems (KAIUS), Key Laboratory of Biomimetic Robots and Systems of Chinese Ministry of Education, School of Mechatronical Engineering, Beijing Institute of Technology, Beijing 100081, China

**Keywords:** teleoperation, dexterous robotic hand, compliant manipulation, adaptive impedance control, fuzzy logic theory, dynamic compliant primitives

## Abstract

Teleoperation enables robots to perform tasks in dangerous or hard-to-reach environments on behalf of humans, but most methods lack operator immersion and compliance during grasping. To significantly enhance the operator’s sense of immersion and achieve more compliant and adaptive grasping of objects, we introduce a novel teleoperation method for dexterous robotic hands. This method integrates finger-to-finger force and vibrotactile feedback based on the Fuzzy Logic-Dynamic Compliant Primitives (FL-DCP) controller. It employs fuzzy logic theory to identify the stiffness of the object being grasped, facilitating more effective manipulation during teleoperated tasks. Utilizing Dynamic Compliant Primitives, the robotic hand implements adaptive impedance control in torque mode based on stiffness identification. Then the immersive bilateral teleoperation system integrates finger-to-finger force and vibrotactile feedback, with real-time force information from the robotic hand continuously transmitted back to the operator to enhance situational awareness and operational judgment. This bidirectional feedback loop increases the success rate of teleoperation and reduces operator fatigue, improving overall performance. Experimental results show that this bio-inspired method outperforms existing approaches in compliance and adaptability during teleoperation grasping tasks. This method mirrors how human naturally modulate muscle stiffness when interacting with different objects, integrating human-like decision-making and precise robotic control to advance teleoperated systems and pave the way for broader applications in remote environments.

## 1. Introduction

The concept of shared control is the human-machine interaction technique that integrates the strengths of both human operators and automated systems to achieve more efficient, safe, and reliable control [1]. In a shared control system, humans and machines do not operate independently but collaborate, leveraging their respective capabilities to jointly accomplish tasks. Shared control aims to achieve better performance by appropriately distributing the control authority between the human operator and the automated system, thereby leveraging both human intelligence and the capabilities of the automation. Shared control is widely applied in fields such as robotics [2], autonomous vehicles [3], teleoperation [4,5,6], and rehabilitation [7,8,9].

With the recent wave of development of humanoid robots, teleoperation technology becomes more and more significant. Teleoperation technology can not only realize the operation when human and robot are separated from each other, so that the robot could replace human to work or explore in dangerous, harsh conditions and hard-to-reach areas; teleoperation technology can also provide high-quality demonstration data for imitation learning and deep reinforcement learning [10,11,12], it endows the robot with ability of autonomous operation, which also accelerates the pace of the robot into the people’s daily life [13].

Teleoperation technology has undergone extensive research and significant development over the past two decades. However, there are still some problems with current hand teleoperation technology. The first problem with most teleoperation systems is their limited immersion, which is either due to the absence of force feedback or the lack of precise finger-to-finger force feedback. Some of them use data gloves without force feedback [14,15,16], and some of them use depth cameras to capture the position of the hand in real time directly [17,18,19]. Although some individuals utilize haptic devices for teleoperation, these systems lack finger-to-finger mapping, let alone precise finger-to-finger force feedback [20,21]. The lack of precise finger-to-finger force feedback makes the human operator subject feel less immersed when grasping the object and will reduce the success rate of teleoperation as a result. The absence of force feedback also makes the subject more susceptible to premature fatigue, as the subject sometimes needs to remain fixed in one position. The aforementioned problems increase the difficulty of novice teleoperation, leading to longer training and adaptation times. The second is the lack of compliance during teleoperation. Most dexterous robotic hands and grippers currently use position control mode to grip objects, which has the advantage of fast response times [22,23,24]. However, even a small deformation of the object after contact can cause excessive gripping force, which may result in damage to the gripped object, irreversible plastic deformation, or mechanical failure of the dexterous robotic hand and gripper. The above problem can be solved by controlling the joint torque to realize impedance control by the dexterous robotic hand [25]. But they have not incorporated targeted variable impedance control specifically adapted to objects with varying stiffness levels. This limitation can result in suboptimal performance when interacting with diverse objects, as the system may not adequately adjust its compliance to match the characteristics of each object. Currently, there are methods that combine machine learning and deep reinforcement learning with variable impedance control [26,27,28,29,30]. But these methods are designed at the end-effectors of robotic arms, which involves simpler forms. In contrast, dexterous hands have five finger end-effectors, resulting in a higher dimensional space, making it difficult to directly transfer these methods to dexterous hands. Additionally, it is challenging to couple and train these methods with teleoperation systems directly.

To address above problems, this paper proposes a novel compliant teleoperation method of hand with finger-to-finger force and vibrotactile feedback based on FL-DCP controller. This method is driven by two primary objectives: the first is to enhance the operator’s sense of immersion during teleoperation, and the second is to achieve more compliant grasping of objects. By providing the operator with more accurate information about the object being grasped at the remote site (such as force feedback and vibrotactile feedback), the sense of immersion during teleoperation can be significantly improved, thereby enhancing the operator’s decision-making capabilities. Compliant grasping of objects with variable stiffness not only protects the object being manipulated, but also safeguards the end-effector. Meanwhile, both of these aspects contribute to an increase in the success rate of teleoperation conducted by the operator and a reduction in operator fatigue. The innovations of this method are listed as follows:This approach pioneeringly combines the fuzzy logic module for estimating the stiffness of grasped objects with the dynamic compliant primitives based self-adaptive regulation of impedance stiffness coefficient, in the context of teleoperated hand manipulation. This integration enables adaptive impedance control that closely mimics human muscle behavior, enhancing the system’s ability to handle objects of different stiffness safely and effectively.The approach integrates compliant adaptive grasping methods into the teleoperation framework for hand manipulation, incorporating finger-to-finger force feedback and vibrotactile feedback on the master side. By integrating force and vibrotactile feedback, the operator gains enhanced situational awareness and operational judgment, leading to better operator confidence and reduced cognitive load. Not only does it increase the success rate of teleoperation and reduce operator fatigue, but it also provides a more immersive and realistic interaction experience.

The remainder of this paper is organized as follows. The methodology is outlined in Section 2. The experimental evaluation is described in Section 3. Finally, the conclusions are presented in Section 4.

## 2. Methodology

### 2.1. Hand Teleoperation Framework Based on Shared Control

The method proposed in this paper falls within the domain of shared control. The hand teleoperation of grasping an object can be divided into two stages, the first stage is when the object has not yet been touched, and the second stage is when the object is touched and the grasping begins. In the first stage, when the dexterous robotic hand is operating in free space, the human operator has full control authority. Then in the second stage, the dexterous robotic hand transitions into operating in constrained space. At this point, the dexterous robotic hand performs object stiffness identification based on FL-DCP and generates desired impedance parameters. Real-time feedback of force and vibrotactile information to the human operator aids in decision-making and enhances the operator’s decision capabilities, providing better expected angles for task completion.

Besides, the proposed method adopts joint torque control mode instead of position control mode to realize compliant grasping. The method utilizes impedance control to regulate the torque output of a dexterous robotic hand involves using the desired position and the actual position as variables. In the first stage, to respond more effectively to the operator’s desired position, the fixed-gain impedance control is adopted. In the second stage, to better adapt to objects with different stiffness, the adaptive impedance control is adopted.

The general framework of the proposed method is depicted in Figure 1. Initially, the data glove with finger-to-finger force feedback and vibrotactile feedback modules on the master side captures the real-time movement angles of the human operator’s fingers. Subsequently, the captured data undergoes cleaning and smoothing processes before being transmitted to the dexterous robotic hand on the follower side. The dexterous robotic hand implements compliant control using impedance control and feedforward control, which operates in joint torque control mode instead of position control mode. The dexterous robotic hand continuously captures its current position and computes the difference with the desired position, which serves as the input for the impedance control. The specifics of impedance control will be elaborated upon in Section 2.3.1. When the force between the dexterous robotic hand and the object exceeds 0.15 N, it is determined that contact has occurred. The dexterous robotic hand subsequently provides real-time force feedback to the master side and transmits tactile vibration signals to notify the operator that contact has been established, transitioning the system into the shared control mode. Upon establishing contact with the object, the dexterous robotic hand identifies the object’s stiffness based on the fuzzy logic module, which will be elaborated upon in Section 2.2. Drawing inspiration from Dynamic Movement Primitives (DMP) and the way humans grasp objects of different stiffness, we propose Dynamic Compliance Primitives (DCP). DCP employ distinct stiffness coefficient profiles tailored to the grasping of objects with varying stiffness levels, which will be elaborated upon in Section 2.3. In the end, the integration and usage of the fuzzy logic module with DCP model to form FL-DCP controller are detailed in Section 2.4.

### 2.2. Fuzzy Logic Module

Human hands are capable of adjusting the grasping stiffness coefficient and grasping force in response to the stiffness of the target object, a process guided by force feedback, tactile feedback and experience. Fuzzy logic theory offers a powerful and flexible approach to control problems characterized by uncertainty, nonlinearity, and complex interactions [31]. Their ability to emulate human reasoning and adapt to changing conditions makes them an attractive choice for many advanced control applications. Therefore, employing a fuzzy logic module to identify and adapt to different stiffness levels objects is a logical choice. Compared to other algorithms, fuzzy logic theory offers two key advantages:It maintains robust performance during input saturation without requiring an extra anti-windup structure [32].It can effectively protect against impulse signals without the need for additional filtering mechanisms.

These features make fuzzy logic theory particularly suitable for applications where reliable and adaptive control is essential [33], such as in robotic grasping tasks.

This paper employs two fuzzy logic controllers, both of which take the grasping force *F* and the grasping force variation rate $\dot{F}$ as input and produce the reference maximum grasping force $F_{r}$ and the reference maximum grasping stiffness coefficient $K_{r}$ as output, respectively. *F* is collected by the force sensor of the dexterous robotic hand in real time. $\dot{F}$ is calculated as(1)$$\dot{F}=\frac{F_{n+1}-F_{n}}{T}$$(2)$$F_{n}=\frac{1}{3}\sum_{i=1}^{3}F_{i}$$
where $F_{i}$ represents the instantaneous value of the contact force between the object and the dexterous robotic hand.

The design details of the fuzzy logic module are as follows, which are shown in Figure 2. Both the input and output membership functions employed standard triangular membership functions to represent linguistic terms, ensuring a structured and intuitive approach to fuzzification and defuzzification processes. The inputs and outputs of the fuzzy logic module are shown in Figure 2. The details of the fuzzy reasoning rules are shown in Table 1, where input variables *F* and $\dot{F}$ are defıned as fıve fuzzy sets: extra small (XS), small (S), medium (M), large (L), and extra large (XL). The output of the fuzzy estimation are the reference grasping stiffness coefficient $K_{r}$ and the reference grasping force $F_{r}$, which are also defıned as fıve fuzzy sets: extra small (XS), small (S), medium (M), large (L), and extra large (XL).

The inference process of fuzzy logic theory utilizes the standard Mandani inference engine. To derive a crisp output, the centroid method is employed, which can be expressed as the following equations.(3)$$K_{r}=\frac{\sum_{m=1}^{M}g^{m}\left[\mu_{A_{i}^{m}}^{*}\left(\alpha_{1}F\right)\mu_{B_{j}^{m}}\left(\alpha_{2}\dot{F}\right)\right]}{\sum_{m=1}^{M}\left[\mu_{A_{i}^{m}}^{*}\left(\alpha_{1}F\right)\mu_{B_{j}^{m}}\left(\alpha_{2}\dot{F}\right)\right]}$$(4)$$F_{r}=\frac{\sum_{m=1}^{M}g^{m}\left[\mu_{A_{i}^{m}}^{*}\left(\beta_{1}F\right)\mu_{B_{j}^{m}}\left(\beta_{2}\dot{F}\right)\right]}{\sum_{m=1}^{M}\left[\mu_{A_{i}^{m}}^{*}\left(\beta_{1}F\right)\mu_{B_{j}^{m}}\left(\beta_{2}\dot{F}\right)\right]}$$

Here, $g^{m}$ denotes the centroid of the consequent set for the $m^{th}$ activated rule, and *M* represents the total number of activated rules. The terms $A_{i}$ and $B_{j}$ denote two input fuzzy sets. The terms $\mu_{A_{i}^{m}}$ and $\mu_{B_{j}^{m}}$ correspond to the membership functions for the inputs *F* and $\dot{F}$. The parameters α and β serve as the coefficients for the inputs, respectively. The symbol * indicates the t-norm operator used to combine these membership values.

### 2.3. Adaptive Impedance Control Based on Dynamic Compliant Primitives

#### 2.3.1. Impedance Control

The dynamics of a rigid body can be represented by the following equation [34,35]:(5)$$M\left(\Theta \right)\ddot{x}+C\left(\Theta ,\dot{\Theta }\right)\dot{x}+G\left(\Theta \right)=\xi_{input}+\xi_{ext}$$Here, Θ and $\dot{\Theta }$ denote the joint angles and velocities, respectively. The variable *x* stands for the position of the end-effector. M(Θ), $C(\Theta ,\dot{\Theta })$ and G(Θ) represent the inertia, the Coriolis and centrifugal forces, and the gravitational force, respectively. $\xi_{input}$ signifies the control input, while $\xi_{ext}$ denotes any external forces acting on the robotic hand, whether from the environment or human interaction.

In accordance with the principles of human muscle motor learning, the control input can be decomposed into two distinct components:(6)$$\xi_{input}=-\xi_{impedance}-\xi_{feedforward}$$In this context, $\xi_{impedance}$ represents impedance component and $\xi_{feedforward}$ represents feedforward component.

The impedance component is typically defined as:(7)$$\xi_{impedance}=K_{s}\varepsilon +K_{d}\dot{\varepsilon }$$
with(8)$$\left.\left\{\begin{array}{c} \varepsilon =q-q_{d}\\ \dot{\varepsilon }=\dot{q}-{\dot{q}}_{d} \end{array}\right.\right.$$Here, $K_{s}$ and $K_{d}$ represent the stiffness coefficient and the damping parameter of impedance control, respectively. ε and $\dot{\varepsilon }$ represent position and velocity errors, respectively. $q_{d}$ and ${\dot{q}}_{d}$ represent the desired position and the desired velocity, respectively.

Inspired by the human process of grasping objects, it is observed that humans exhibit a variable stiffness behavior during the act of grasping [36]. This characteristic is crucial for performing delicate and complex tasks, such as handling objects with high deformability or performing precise operations. When humans grasp objects of different shapes, sizes, and textures, the muscles of their hands adjust their stiffness coefficient to suit the specific requirements of the grip. Therefore, teleoperated dexterous robotic hand grasping objects can be divided into two stages, the first stage is when the object has not yet been touched, and the second stage is when the object is touched and the grasping begins. In the first stage fixed gain impedance control is used and in the second stage adaptive impedance control is used.

#### 2.3.2. The Definition and Derivation of Dynamic Compliant Primitives

Dynamic Movement Primitives (DMP) is widely used for the demonstration and generalization of robot trajectory [37]. Just as humans possess the ability to adapt stiffness coefficient during grasp interactions, robots should similarly be capable of learning and generalizing highly human-like skills. Stiffness coefficient adaption refers to the process of dynamically adjusting the stiffness coefficient applied to an object during grasping, based on the object’s stiffness characteristics. Although human hands use different stiffness coefficient settings when handling objects with varying stiffness, these adjustments follow a common trend. This phenomenon provides a foundation for developing dexterous robotic hand grasping algorithms that can automatically adapt to objects of different stiffness. To achieve this, it is crucial to place significant emphasis on the representation of stiffness coefficient profiles. Furthermore, the DMP framework can be extended to the Dynamic Compliant Primitives (DCP) framework for stiffness coefficient adaptation. Based on the DMP framework, the DCP framework can be utilized to learn and generalize the trajectory of stiffness coefficient profiles. The definition and derivation of DCP are presented as follows.

Similarly to DMP models, DCP models can generally be categorized into two types based on the nature of the tasks: discrete type and rhythmic type [38,39]. This paper employs the former type. The DCP model is conceptualized as a spring damper system plus an external force component, which originates from a standard dynamical system. The specifics of this model are outlined below:(9)$$\begin{array}{cc} \lambda \dot{z} & =k_{s}\left(x_{t}-x\right)-k_{d}\dot{x}+\left(x_{t}-x_{0}\right)f\left(s\right) \end{array}$$(10)$$\begin{array}{cc} \lambda \dot{x} & =z \end{array}$$In this scenario, *x* denotes the stiffness coefficient profiles, while *z* represents the velocity of change in the stiffness coefficient profile. The symbols $x_{0}$ and $x_{t}$ indicate the starting and target stiffness coefficient values, respectively. The parameters $k_{s}$ and $k_{d}$ represent the stiffness and damping coefficients of the spring damper system, respectively. The constant λ is positive, which serves as the speed modulation parameter. Upon adjustment of the coefficient λ, the time required to complete a task can be modified. The non-linear function f(s) can be modeled as a linear combination of radial basis functions, capturing its complex behavior in a structured manner. The variable *s* reflects the state of the first order dynamical system that has the same speed modulation parameter λ. The evolution of *s* is governed by the following differential equation:(11)$$\lambda \dot{s}=-\alpha_{s}s$$
where $\alpha_{s}$ is a predetermined constant and s∈[0,1].

In this system, regardless of the initial value, *s* ultimately converges monotonically to the target state as time approaches infinity, thus it could be regarded as a phase parameter. And *s* exhibits monotonically increasing behavior throughout its entire domain. As *s* approaches 0, f(s) decreases, reaching 0 at the target stiffness coefficient value. Owing to the characteristics of the standard system, the DCP model utilizes the phase parameter *s* rather than time as its basis, allowing for easy adaptation to various scenarios without altering the trajectory of the stiffness coefficient profile. The two parameters $\alpha_{s}$ and λ are able to influence the system’s convergence rate.

The external force term f(s) is defined by(12)$$f\left(s\right)=\sum_{i=1}^{N}w_{i}\phi_{i}\left(s\right)s$$
with(13)$$\phi_{i}\left(s\right)=\frac{\exp(-h_{i}{\left(s-c_{i}\right)}^{2})}{\sum_{j=1}^{N}\exp\left(-h_{j}{\left(s-c_{j}\right)}^{2}\right)}$$The normalized radial basis function $\phi_{i}\left(s\right)$, characterized by its center $c_{i}$ and bandwidth $h_{i}$, is employed in the model. Each of these basis functions is associated with a weight $w_{i}$. The total number of Gaussian functions utilized within this framework, denoted as *N*, is determined based on the specific requirements of the task at hand. Parameters such as the center $c_{i}$, the bandwidth $h_{i}$, and the quantity *N* are established according to the specific scenario and tailored to the specific requirements of impedance control.

Given a single stiffness coefficient profile demonstration $\left[x_{demo},{\dot{x}}_{demo},{\ddot{x}}_{demo}\right]$, the target nonlinear function $f_{target}\left(s\right)$ can be represented as follows:(14)$$f_{target}\left(s\right)=\frac{\lambda^{2}{\ddot{x}}_{demo}+k_{d}{\dot{x}}_{demo}-k_{s}\left(x_{t}-x_{demo}\right)}{x_{t}-x_{0}}$$Then, the weight *w* of DCP can be calculated using LWR (locally weighted regression) method by minimizing the cost function. The cost function is described as follows:(15)$$J=\sum {\left(f_{target}\left(s\right)-f\left(s\right)\right)}^{2}$$

By learning the specific pattern of how stiffness coefficient varies with independent variables from a concrete example, DCP model are able to capture and encode the key features in this process. Once such a model is established, it is capable of generating appropriately adjusted stiffness coefficient control strategies when confronted with new target objects having different stiffness properties. This approach not only preserves the fundamental behavioral logic embodied in the original demonstration data but also demonstrates adaptability to unknown environmental conditions.

The simulation results of the DCP model, shown in Figure 3, demonstrate adaptive reproduction of stiffness coefficient profiles under varied initial and final conditions. From Figure 3, it is evident that once a stiffness coefficient profile has been established for learning purposes, the system demonstrates remarkable adaptability in reproducing similar contours across subsequent trials. Lines 1 to 5 are the reproduced contours. Specifically, even when the starting point remains constant but the endpoint varies, or when both the starting and ending points differ from the original demonstration, the system is capable of accurately generating a stiffness coefficient profile that closely resembles the initially demonstrated contour. It demonstrates its versatility and reliability in achieving compliant and precise grasping behaviors.

### 2.4. Fuzzy Logic-Dynamic Compliant Primitives Controller

To achieve variable stiffness coefficient control that closely emulates the adaptability of the human hand during object grasping, we propose an advanced FL-DCP control framework. This system is designed to dynamically adjust the stiffness coefficient of a robotic hand in response to varying object properties and interaction forces, ensuring compliance in manipulation tasks.

During this learning process, the system captures the nuances of the desired stiffness coefficient variation, which can be later reproduced with high fidelity. The learned stiffness coefficient profile serves as a template for subsequent operations, guiding the robotic hand’s response to different grasping scenarios. The initial phase of this control strategy involves learning a stiffness coefficient profile that can smoothly transition between a fixed initial stiffness coefficient $K_{0}$ and a fixed final stiffness coefficient $K_{e}$. To ensure that this transition is continuous and differentiable, thereby avoiding abrupt changes that could lead to instability, we employ the Sigmoid function [40]. The Sigmoid function allows for a smooth interpolation between the starting point and the end point of the trajectory, providing a natural progression that mimics human muscle behavior when adapting to the objects being manipulated.

Once the stiffness coefficient profile has been established, the system enters the reproduction phase. Here, the output of the fuzzy logic module provides reference values for grasping stiffness coefficient $K_{r}$ and force $F_{r}$, which are critical parameters to achieving the intended level of compliance. The stiffness coefficient profile begins at the same fixed initial stiffness coefficient $K_{0}$ but now transitions to the reference grasping stiffness coefficient $K_{r}$, rather than the previously fixed final stiffness coefficient $K_{e}$. This adjustment is crucial because it allows the system to respond more flexibly to the specific requirements of each grasping task. By aligning the final grasping stiffness coefficient value with the reference grasping stiffness coefficient derived from the fuzzy logic module, the controller can better accommodate the unique properties of different object.

Central to the DCP model is the input variable *R*, which represents the ratio of the current contact force $F_{c}$ to the reference grasping force $F_{r}$:(16)$$R=\frac{F_{c}}{F_{r}}$$Here, $F_{c}$ is defined as:(17)$$F_{c}=\left\{\begin{array}{c} F_{c},\;F_{c}\leq F_{r}\\ F_{r},\;F_{c}>F_{r} \end{array}\right.$$This ensures that the applied force does not exceed the reference value, preventing potential damage to either the object or the robotic hand. The output of the DCP model is the adapted stiffness coefficient $K_{s}$, which plays a pivotal role in the impedance control. The adapted stiffness coefficient $K_{s}$ is continuously adjusted based on the input ratio *R*, allowing the robotic hand to maintain adequate compliance while interacting with various objects.

By integrating these components within the FL-DCP control framework, we create a robust system capable of performing compliant and adaptive grasping tasks with accuracy and responsiveness, closely mirroring the sophisticated capabilities of the human hand.

## 3. Experimental Evaluation

To verify the effectiveness of the proposed method, three distinct hand teleoperation experiments are carried out. The first experiment is conducted to validate the efficacy of the object stiffness identification method based on the fuzzy logic module. The second experiment aimed to demonstrate that this adaptive impedance control method results in more compliant grasping than alternative approaches. The third experiment aimed to demonstrate that integrating this adaptive impedance control method with hand teleoperation enhances the success rate of teleoperation tasks and reduces operator fatigue. In above experiments, the motion generator used on the master side is Senseglove Nova 2, which is equipped with force feedback module and vibrotactile feedback module. And the dexterous robot hand used on the follower side is Inspire Hand, which is equipped with force and position sensors.

The specific teleoperation details are as follows. On the master side, Senseglove Nova 2 is connected to a Windows computer via Bluetooth. The data glove captures the human operator’s finger angles in real-time. After being processed with the sliding window mean filter and scale transformation, the data are transmitted over a local network using the TCP/IP protocol. On the follower side, Inspire Hand is connected to a Linux computer. Similarly, the dexterous robotic hand receives the desired angle data via the TCP/IP protocol and sends the force feedback data in real-time. When the data glove receives force data from the dexterous robotic hand, it applies this force feedback directly to the operator’s hand, providing a realistic sense of touch. If the detected force exceeds 0.15 N, the data glove activates its vibrotactile feedback module. This vibration serves as an immediate notification to the operator that the dexterous robotic hand has come into contact with the object.

### 3.1. Object Stiffness Identification

To validate the efficacy of the object stiffness identification method based on the fuzzy logic module, five common objects with different stiffness levels were chosen for the experiment. The objects, ordered from highest to lowest stiffness, are: hard metal bottle, hard plastic bottle, medium plastic bottle, soft plastic bottle, and soft sponge.

At first, we used the apparatus shown in Figure 4 to measure the actual stiffness of different objects. To determine the actual stiffness of the objects, we quantify their deformation in response to an applied force. This process involves using a vernier caliper to precisely measure the object’s deformation and a force sensor to accurately record the applied force. The actual measured stiffness of the objects is displayed in the scatter graph shown in Figure 5b, with each object tested 5 times repeatedly. It can be observed that the stiffness of the five objects can be categorized as very hard, relatively hard, medium, relatively soft, and very soft. For very hard objects, we assume that they do not deform, and their stiffness is considered to be infinitely large. Due to the exceedingly high stiffness of the very hard object and the consequent difficulty in obtaining precise measurements, the data points representing the very hard object are not depicted in Figure 5b. This decision was made to enhance the readability of the graph and the visual differentiation of stiffness among other objects. Additionally, it is noted that objects with higher stiffness exhibit larger variances in their measured stiffness values.

The experimental process of object stiffness identification is as follows: By setting the same finger torque, the dexterous robotic hand contacts objects with different stiffness levels, ensuring that contact position between the objects and the robotic hand remains as consistent as possible. The results of the stiffness identification are recorded and analyzed.

The force variation experienced by the index finger of the dexterous robotic hand during the contact with the object is illustrated in Figure 5a. When the force exceeds 0.15 N, it is determined that the dexterous robotic hand has initiated contact with the object, and the stiffness identification process begins. The data collection for the stiffness identification process continues for 1 s, as indicated by the red region in Figure 5a. During this period, the system gathers force data to identify the stiffness of the object. After collecting the force information, the rate of variation of the force is computed. The force and its rate of variation are then used as inputs to the fuzzy logic module.

The output results of the fuzzy logic module are presented in the line graph in Figure 5b, including the maximum reference force and the maximum grasping reference stiffness coefficient, which are represented by blue squares and purple hexagons, respectively. It is important to note that the grasping stiffness coefficient referred to here is the stiffness coefficient parameter $K_{s}$ used in the impedance control law. In this case, the fuzzy logic module outputs the maximum set values for the reference force and reference grasping stiffness coefficient. The output range for the maximum reference force is limited to 1–4 N, and the output range for the maximum grasping stiffness coefficient is limited to 1–5. Therefore, for very hard objects, the fuzzy logic module outputs the maximum reference force of 4 N and the maximum grasping stiffness coefficient of 5. Besides, it can be observed that for objects with higher stiffness, both the maximum reference force and the maximum reference grasping stiffness coefficient are higher. Conversely, for objects with lower stiffness, both the maximum reference force and the maximum reference grasping stiffness coefficient are lower. Furthermore, it is observed that the trends in the maximum reference force and the maximum reference grasping stiffness coefficient closely follow the trends in the stiffness of the objects. This finding essentially confirms the effectiveness of the object stiffness identification method based on the fuzzy logic module.

### 3.2. Comparison of Different Control Methods for the Dexterous Robotic Hand

To validate the efficacy of the proposed adaptive impedance control, three different control methods for the dexterous robotic hand were compared. Three different control methods are position control mode, fixed-gain impedance control, and adaptive impedance control, respectively.

The experimental process is as follows: Three different control methods were used to teleoperate the dexterous robotic hand to grasp objects with three different levels of stiffness. The three objects are soft sponge, soft plastic bottle and hard plastic bottle. The contact force was recorded once the grasp stabilized. Each grasping test was repeated 10 times and the mean and standard deviation of contact forces were calculated for analysis.

The contact force data collected during experiments conducted in adaptive impedance mode is illustrated in Figure 6a. As illustrated in Figure 6a, it can be observed that as the object transitions from soft to hard, the contact force increases progressively from the initial grasping phase until stabilization. Additionally, the rate at which the contact force changes becomes faster with increasing object stiffness. The experimental result of different control methods for the dexterous robotic hand is illustrated in Figure 6b and Table 2. From Figure 6b, it can be observed that the contact force exerted on the objects of different stiffness levels varies significantly depending on the control method used. The contact force of position control is the highest among the three control methods. This is likely due to the rigid nature of position control, which does not account for the compliance of the objects, leading to higher forces being applied. The contact force of fixed gain impedance control is moderate, indicating that impedance control provides a balance between precision and compliance, allowing for safer interaction with the objects. The contact force of adaptive impedance control is the lowest among the three control modes. This is a significant advantage of the adaptive impedance control proposed in this paper, as it dynamically adjusts the stiffness coefficient and force profiles based on the object’s stiffness, resulting in minimal force application and safer handling of the objects. Simultaneously, it can be observed that the standard deviation of the contact force under position control is much larger compared to impedance control modes. And the standard deviations of the contact forces for the two impedance control modes are relatively similar, indicating similar stability and consistency. This is because, under position control mode, even small changes in the desired position can lead to substantial variations in the contact force. The experimental results presented above effectively validate the effectiveness of the adaptive impedance control method proposed in this paper for compliantly grasping objects with different stiffness levels.

### 3.3. Robotic Arm and Hand Teleoperation Experiment

To validate the effectiveness of the proposed compliant hand teleoperation method, we applied it to actual teleoperation scenarios and conducted experiments. The general framework for teleoperated robotic hand and arm is shown in Figure 7. The experimental platform utilized in teleoperation experiments is the immersive humanoid dual-arm dexterous robot previously developed by our laboratory team [41]. The data glove and motion capture use Senseglove Nova 2 and Noitom Perception Neuron Pro on the master side, respectively. The dexterous robotic hand and the robotic arm use Inspire Hand and Rokae xMate ER7 Pro on the follower side, respectively. The data communication between the motion capture system and the robotic arm is also achieved via TCP/IP protocol over a local area network. The motion capture data is divided into position and orientation components, which are transmitted in incremental form for action mapping. After receiving the pose data, the robotic arm uses inverse kinematics to calculate the appropriate joint angles to move. The inverse kinematics adopts the relaxed-ik method [42]. This method can avoid kinematics singularity, discontinuity of joint space, self-collision avoidance and so on while ensuring the end-effector pose matching.

The scenarios of arm and hand teleoperation experiment are shown in Figure 8. The experimental process consists of two parts. In the first part, the teleoperated dexterous robotic hand and robotic arm are used to grasp and suspend objects until the operator’s hand feels fatigued. The fatigue time is tested and recorded under two conditions: with or without force feedback and vibrotactile feedback. Each condition is repeated 10 times. The average fatigue time and standard deviation are calculated. In the second part, the success rate of teleoperated grasping and placing objects at another location is evaluated using five objects differ in stiffness from the previous experiment. The testing is performed under four conditions: with or without force feedback and vibrotactile feedback, and in position control mode or adaptive impedance mode. Each object is grasped and placed 5 times, and the success rate is recorded. A grasp is considered a failure if the object drops or the force exceeds the maximum reference grasping force established in the Section 3.1 of the experimental evaluation. The average success rate across all five objects is then calculated.

The experimental result is shown in Table 3 and Table 4. It can be observed that the presence of force feedback and vibrotactile feedback during teleoperation significantly increases the operator’s fatigue time and operational success rate compared to conditions without feedback, demonstrating the necessity of force feedback and vibrotactile feedback in teleoperation. The adaptive impedance mode, compared to position control mode, can improve the success rate of teleoperation and help the operator perform tasks more easily. In summary, the use of adaptive impedance mode, along with force feedback and vibrotactile feedback, is the optimal choice for teleoperated hand manipulation. Furthermore, these findings also validate the effectiveness of the method proposed in this paper.

Besides, we tested both the communication time over TCP/IP protocol and the processing time of the control algorithm. The experimental results are as follows: under optimal network conditions, the average transmission latency measured over TCP/IP protocol was 4.56 ms, with a maximum observed latency of 8.58 ms. The average processing time for controlling the dexterous hand directly using the position control method was 31.9 ms, while using the method proposed in this article, the average processing time was 54.6 ms. Although the processing time of the method proposed in this article is longer compared to the directly position control method, the difference in operational experience during tasks that do not require high-speed operation is not substantial. In addition to the processing speed being slower compared to the direct position control method, another limitation is that the dexterous hand’s response time is less sensitive. Specifically, the responsiveness of the dexterous hand of our proposed method is not as immediate as it is with the position control method.

## 4. Conclusions

In this study, we introduce an innovative compliant teleoperation method of hand that significantly enhances the operator’s sense of immersion and achieves more compliant grasping of objects. This method integrates finger-to-finger force feedback and vibrotactile feedback based on the FL-DCP controller, which fully leverages the decision-making capabilities of both humans and robots. The dexterous robotic hand operates in torque mode to perform impedance control, allowing for precise and adaptive interactions with objects.

A key feature of our approach is the implementation of adaptive impedance control by the FL-DCP controller, which identifies the stiffness of the grasped object and adjusts its response accordingly. This mimics the human ability to modulate muscle stiffness coefficient when interacting with different objects, leading to more intuitive and effective manipulation during teleoperated tasks. This bio-inspired strategy enables more compliant grasping of different stiffness objects. To further enhance the operator’s performance, real-time force information from the robotic hand is transmitted back to the operator, thereby improving judgment and reducing fatigue during extended periods of teleoperation. This bidirectional communication loop ensures that the operator remains well-informed about the interaction forces at play, contributing to higher success rates and less fatigue in grasping tasks. Our method has been validated through stiffness identification and practical teleoperation experiments, demonstrating superior performance compared to existing methods. Overall, this novel compliant teleoperation method of hand represents a significant advancement in the field, offering a promising direction for improving human-robot collaboration in remote manipulation tasks.

The potential future work includes the incorporation of tactile information from the robotic fingers into the system. This addition aims to provide even more nuanced feedback, potentially leading to even more compliant and sensitive grasping capabilities. The second direction is to investigate the impact of latency on the controller and develop strategies to mitigate its effects on the operator’s experience. And another potential direction would be to use machine learning-based controllers to model how humans vary impedance coefficients during the grasping of different objects and then apply these learned patterns to impedance control.

## Figures and Tables

**Figure 1 biomimetics-10-00194-f001:**
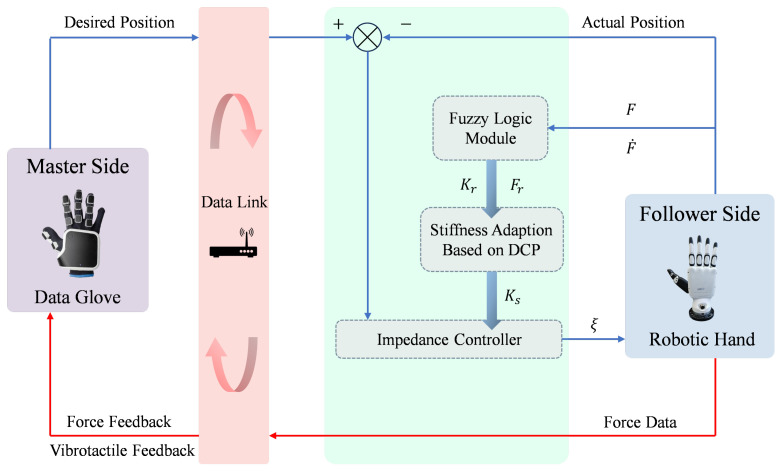
General framework of hand teleoperation.

**Figure 2 biomimetics-10-00194-f002:**
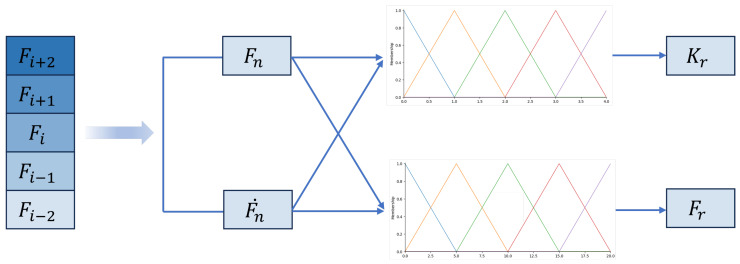
Inputs and outputs of fuzzy logic module.

**Figure 3 biomimetics-10-00194-f003:**
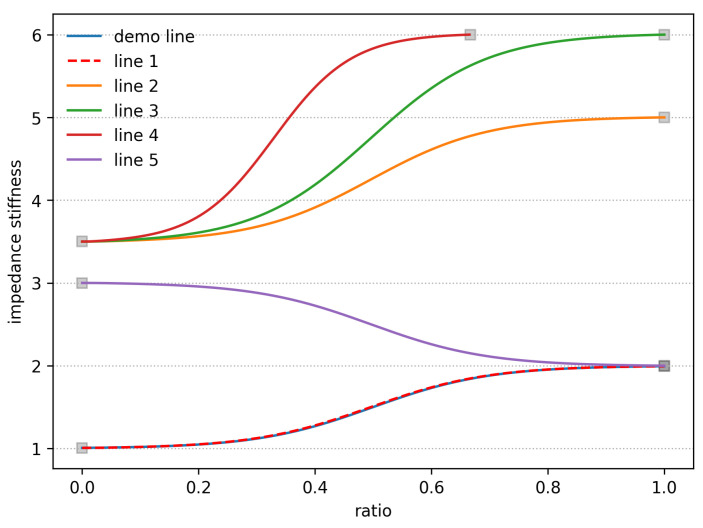
Adaptive reproduction of stiffness coefficient profiles under varied initial and final conditions.

**Figure 4 biomimetics-10-00194-f004:**
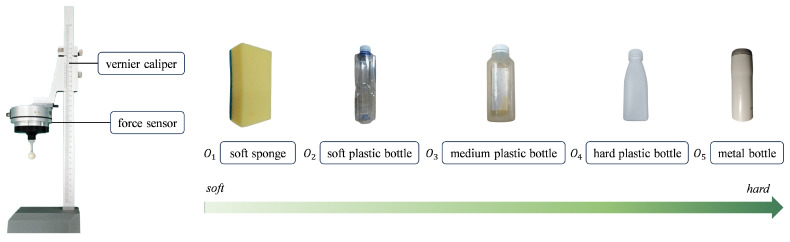
The experimental apparatus of the object stiffness identification.

**Figure 5 biomimetics-10-00194-f005:**
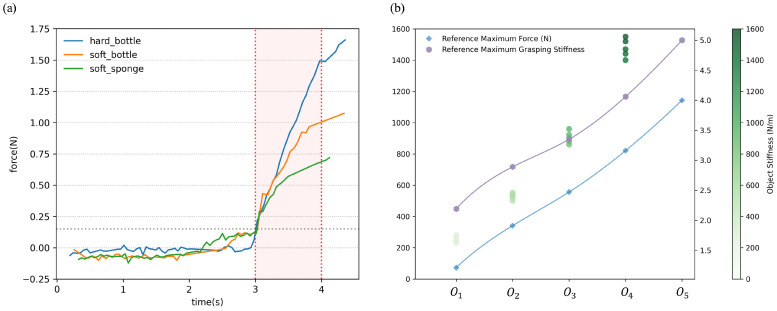
(**a**) The force data collected during the object stiffness identification experiment. (**b**) The result of the object stiffness identification experiment.

**Figure 6 biomimetics-10-00194-f006:**
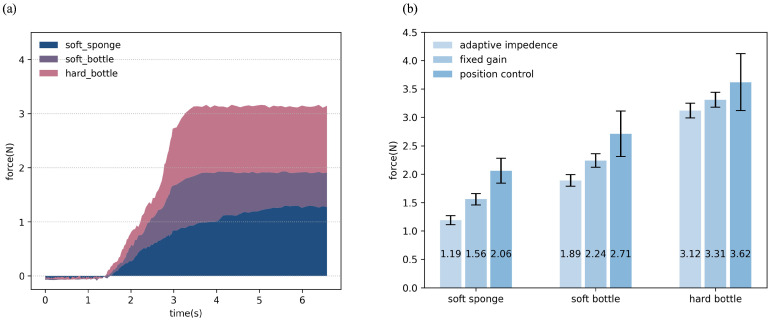
(**a**) The force data collected during experiments conducted in adaptive impedance mode. (**b**) The experimental result of different control methods for the dexterous robotic hand.

**Figure 7 biomimetics-10-00194-f007:**
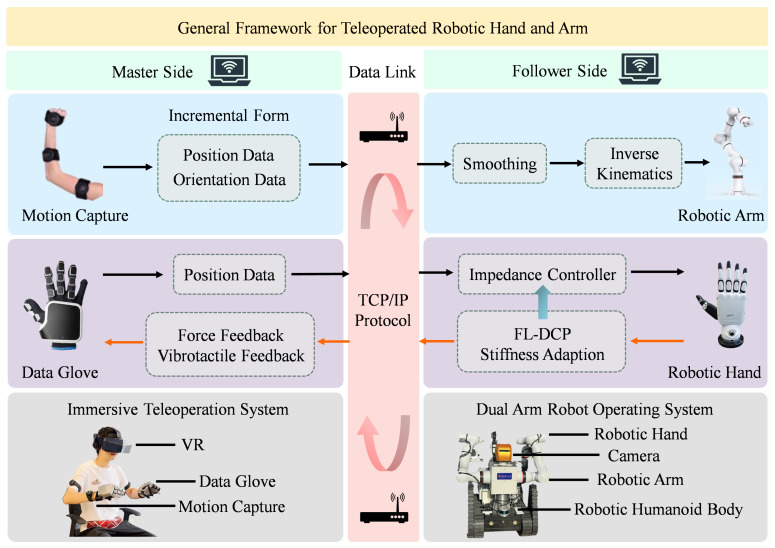
General framework for teleoperated robotic hand and arm.

**Figure 8 biomimetics-10-00194-f008:**
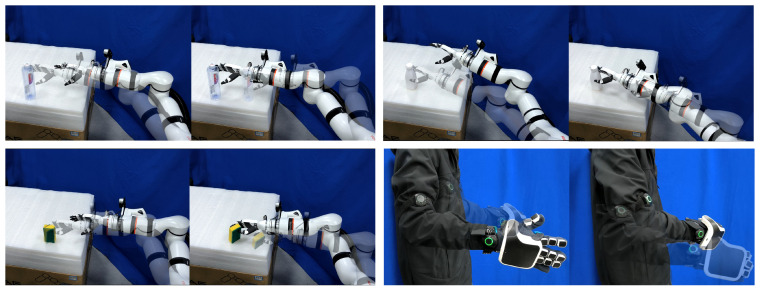
The scenarios of arm and hand teleoperation experiment.

**Table 1 biomimetics-10-00194-t001:** Fuzzy reasoning rules.

		$\dot{F}$				
		XL	L	M	S	XS
F	**XL**	XL	XL	L	M	S
	**L**	XL	L	M	M	S
	**M**	L	L	M	S	S
	**S**	L	M	M	S	XS
	**XS**	L	M	S	XS	XS

**Table 2 biomimetics-10-00194-t002:** The experimental result of different control methods for the dexterous robotic hand.

	Soft Sponge	Soft Bottle	Hard Bottle
position control	2.06 ± 0.22	2.71 ± 0.41	3.62 ± 0.52
fixed gain	1.56 ± 0.09	2.24 ± 0.12	3.31 ± 0.13
adaptive impedance	1.19 ± 0.08	1.89 ± 0.10	3.12 ± 0.13

**Table 3 biomimetics-10-00194-t003:** The result of the teleoperation experiment about fatigue time.

	Fatigue Time (s)
without force and vibrotactile feedback	21.9 ± 2.2
with force and vibrotactile feedback	46.4 ± 3.6

**Table 4 biomimetics-10-00194-t004:** The result of the teleoperation experiment about success rate.

	Success Rate
without feedback & position control	52%
without feedback & adaptive impedence	64%
with feedback & position control	68%
with feedback & adaptive impedence	82%

## Data Availability

The data presented in this study are available on request.

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
