# Peer review of "A Novel Hand Teleoperation Method with Force and Vibrotactile Feedback Based on Dynamic Compliant Primitives Controller"

_biomimetics, 2025, doi:10.3390/biomimetics10040194_

Round 1
Reviewer 1 Report
Comments and Suggestions for Authors
Comments:
1) The abstract lacks clarity and completeness. It does not sufficiently explain the need for the study, the key findings, or the broader implications of the research. Please revise the abstract to include the motivation behind the study, the research gap addressed, the methodology, key outcomes, and their significance.
2) In the introduction, the authors should provide a detailed comparison with other state-of-the-art methods, such as machine learning-based controllers and deep reinforcement learning approaches, beyond basic impedance control methods. Therefore, the introduction should be revised accordingly.
3) Since force feedback and vibrotactile feedback rely on real-time data transmission, the authors should provide a more detailed discussion on latency, processing time, and potential delays.
4) The authors should provide more details on how communication delays affect user experience and operational accuracy.
5) The authors claim that the system reduces operator fatigue, but they do not provide a direct quantitative analysis of how quickly users adapt to the new feedback mechanism compared to traditional teleoperation.
6) While the results are promising, it would be beneficial to include comparisons with a broader range of existing methods, including those without force feedback. This would provide a clearer context for the innovation relative to other approaches. In other words, the authors should perform a comparative analysis and emphasize the advantages of the proposed method over existing solutions.
7) I encourage the authors to compare the proposed system with machine learning-based teleoperation frameworks or data-driven adaptive controllers.
8) What are the limitations of the proposed approach? A more detailed analysis and discussion on this topic are necessary. I would also appreciate an open and honest discussion of its limitations.
9) The relative work in the following reference can be mentioned in the introduction: https://doi.org/10.1007/s13369-023-07720-0
Author Response
We appreciate the time and effort you dedicated to providing feedback on our manuscript and are grateful for the insightful comments and valuable improvements to our paper. We have incorporated most of the reviewers’ suggestions, which are in red within the revised manuscript. Please see below, for a point-by-point response to the reviewers’ comments and concerns.
Comments 1:
The abstract lacks clarity and completeness. It does not sufficiently explain the need for the study, the key findings, or the broader implications of the research. Please revise the abstract to include the motivation behind the study, the research gap addressed, the methodology, key outcomes, and their significance.
Response 1:
Thank you for your valuable suggestion. We have revised the abstract as shown below according to your suggestion. The revised abstract includes the motivation behind the study, the research gap addressed, the methodology, key outcomes, and their significance.
The motivation behind the study: Teleoperation enables robots to perform tasks in dangerous or hard-to-reach environments on behalf of humans, but most methods lack operator immersion and compliance during grasping.
The research gap addressed: To significantly enhance the operator’s sense of immersion and achieve more compliant and adaptive grasping of objects, we introduce a novel teleoperation method for dexterous robotic hands.
The methodology: This method integrates finger-to-finger force feedback and vibrotactile feedback based on the Fuzzy Logic-Dynamic Compliant Primitives (FL-DCP) controller. It employs fuzzy logic theory to identify the stiffness of the object being grasped, facilitating more effective manipulation during teleoperated tasks. Utilizing Dynamic Compliant Primitives, the robotic hand implements adaptive impedance control in torque mode based on stiffness identification. Then the immersive bilateral teleoperation system integrates finger-to-finger force and vibrotactile feedback, with real-time force information from the robotic hand continuously transmitted back to the operator to enhance situational awareness and operational judgment.
Key outcomes: This bidirectional feedback loop increases the success rate of teleoperation and reduces operator fatigue, improving overall performance. Experimental results show that this bio-inspired method outperforms existing approaches in compliance and adaptability during teleoperation grasping tasks.
Significance: This method mirrors how human naturally modulate muscle stiffness when interacting with different objects, integrating human-like decision-making and precise robotic control to advance teleoperated systems and pave the way for broader applications in remote environments.
Comments 2:
In the introduction, the authors should provide a detailed comparison with other state-of-the-art methods, such as machine learning-based controllers and deep reinforcement learning approaches, beyond basic impedance control methods. Therefore, the introduction should be revised accordingly.
Response 2:
Thank you for your valuable suggestion. Admittedly, machine learning-based controllers and deep reinforcement learning approaches are highly effective methodologies. Currently, there are methods that combine machine learning and deep reinforcement learning with variable impedance control [26-30]. But these methods are designed at the end-effectors of robotic arms, which involves simpler forms. In contrast, there are virtually no applications that combine reinforcement learning with variable impedance control for dexterous hands. Because dexterous hands have five finger end-effectors, resulting in a higher dimensional space, making it difficult to directly transfer these methods to dexterous hands. Additionally, it is challenging to couple and train these methods with teleoperation systems directly. This paper primarily focuses on enhancing compliance and immersion during the teleoperation of dexterous hands. It is added to lines 62-67 of the revised manuscript with track changes.
If machine learning-based controllers and deep reinforcement learning approaches methods are to be applied to hand teleoperation, a potential direction would be to use learning approaches to model how humans vary impedance coefficients during the grasping of different objects and then apply these learned patterns to impedance control. Besides, this represents a future direction for our research. It is added to lines 498-501 of the revised manuscript with track changes.
Comments 3:
Since force feedback and vibrotactile feedback rely on real-time data transmission, the authors should provide a more detailed discussion on latency, processing time, and potential delays.
Response 3:
Thank you for pointing this out. Latency is indeed crucial for teleoperation. Therefore, we tested both the communication time over TCP/IP protocol and the processing time of the method.
The experimental results are as follows: under optimal network conditions, the average transmission latency measured over TCP/IP protocol was 4.56ms, with a maximum observed latency of 8.58ms. The average processing time for controlling the dexterous hand directly using the position control method was 31.9ms, while using the method proposed in this article, the average processing time was 54.6ms. The result is added to lines 458-466 of the revised manuscript with track changes.
Comments 4:
The authors should provide more details on how communication delays affect user experience and operational accuracy.
Response 4:
Thank you for your valuable suggestion. Communication delays can indeed affect the user experience and operational accuracy. The proposed method introduces a latency of approximately 50-60ms. During tasks that do not require high-speed operation, this latency has a minimal impact, and operators are unlikely to perceive any significant effects. However, from a theoretical standpoint, latency can still influence user experience and operational accuracy. To address this, we plan to conduct further research focused on mitigating the impact of latency on the controller. And the potential future work is added to the lines 496-498 of the revised manuscript with track changes.
Comments 5:
The authors claim that the system reduces operator fatigue, but they do not provide a direct quantitative analysis of how quickly users adapt to the new feedback mechanism compared to traditional teleoperation.
Response 5:
Thank you for your valuable suggestion. This teleoperation system is relatively simple and intuitive. To ensure that all operators are novices, there is no specialized learning or adaptation phase. At the start of the testing, operators are informed about the operation procedures. Under these conditions, testing success rates and fatigue time can reasonably validate the effectiveness of our method.
In this article, reducing operator fatigue refers to decreasing the physical strain experienced by the operator's hand during the teleoperation process. This fatigue typically occurs because the operator's hand must maintain a specific posture for extended periods while grasping objects. It does not refer to how quickly an operator can adapt to the new feedback mechanism. By integrating finger-to-finger force feedback and vibrotactile feedback into our control framework, operators can more easily maintain a specific posture, and the sense of immersion during teleoperation is also increased. And quantitative analysis of fatigue time is shown in the Table 3.
Comments 6:
While the results are promising, it would be beneficial to include comparisons with a broader range of existing methods, including those without force feedback. This would provide a clearer context for the innovation relative to other approaches. In other words, the authors should perform a comparative analysis and emphasize the advantages of the proposed method over existing solutions.
Response 6:
Thank you for your valuable suggestion. Currently, most existing variable impedance control methods are designed for use with the end-effectors of robotic arms. In contrast, the variable impedance approach discussed in this paper is tailored for dexterous hands, which complicates direct comparisons between the two applications. For dexterous hands, traditional methods generally involve position control [22-24], which aligns with the existing solutions used in our experimental setup.
In the article, we conducted three sets of comparative experiments. The first set involved using three different control methods to grasp objects of different stiffness and comparing the forces exerted during these grasps. The three control methods were: dexterous hand position control, traditional fixed-gain impedance control, and adaptive impedance control based on FL-DCP. The experimental results presented in Table 2 demonstrate the advantages and innovations of our proposed method in terms of compliance compared with other methods. The second set of experiments compared the performance with and without force feedback and vibrotactile feedback, using the same control method. Specifically, we evaluated how these feedback mechanisms affect operator fatigue during teleoperation tasks. The experimental results, detailed in Table 3, clearly show the advantages of feedback mechanisms in reducing operator fatigue. Our findings indicate that incorporating force feedback and vibrotactile feedback significantly enhances the operator's comfort by making it easier to maintain specific postures for extended periods, thereby improving overall task performance and user experience. The third set of experiments compared the success rates of teleoperation tasks under different combinations of control methods and feedback mechanisms. The experimental results presented in Table 4 demonstrate that the combination of adaptive impedance control based on FL-DCP with feedback achieves the highest success rates in teleoperation tasks, which validate the superiority of the proposed method in terms of compliance and immersion.
Comments 7:
I encourage the authors to compare the proposed system with machine learning-based teleoperation frameworks or data-driven adaptive controllers.
Response 7:
Thank you for your valuable suggestion. Machine learning-based teleoperation frameworks and data-driven adaptive controllers are highly effective, but current machine learning and data-driven adaptive controllers are predominantly tailored for end-effector operations of robotic arms. In contrast, this paper focuses on variable impedance control within dexterous hands, which complicates direct comparisons with existing methods. Despite this, Machine learning-based teleoperation frameworks and data-driven adaptive controllers hold significant promise due to their ability to reduce reliance on detailed models. Consequently, integrating these methods into dexterous hand teleoperation will be a key focus of our future research. The potential direction would be to use machine learning-based or data-driven adaptive controllers to model how humans vary impedance coefficients during the grasping of different objects and then apply these learned patterns to impedance control. And the potential future work is added to the lines 498-501 of the revised manuscript with track changes.
Comments 8:
What are the limitations of the proposed approach? A more detailed analysis and discussion on this topic are necessary. I would also appreciate an open and honest discussion of its limitations.
Response 8:
Thank you for your valuable suggestion. The first limitation of the proposed approach is the processing time of the method proposed in this article is longer compared to the direct position control method. In addition to the processing speed being slower compared to the direct position control method, another limitation is that the dexterous hand's response time is less sensitive. And the limitations of the proposed approach are added to the lines 458-470 of the revised manuscript with track changes.
Comments 9:
The relative work in the following reference can be mentioned in the introduction: https://doi.org/10.1007/s13369-023-07720-0.
Response 9:
Thank you for your valuable suggestion. https://doi.org/10.1007/s13369-023-07720-0 is mentioned in the introduction and added to the references.
Reviewer 2 Report
Comments and Suggestions for Authors
Comment 1:
Quoted: “Shared control is widely applied in fields such as robotics [2], autonomous vehicles [3], teleoperation [4–6], and assistive technologies [7].”
Comment: Shared control is part of robotics. Therefore, the application of shared control should include rehabilitation, e.g., ‘Design and Rapid Construction of a Cost-Effective Virtual Haptic Device. ‘
Comment 2:
Quoted: However, there are still some problems with current hand teleoperation technology, the first is that most hand teleoperation is without force feedback, some of them use data gloves without force feedback.”
Comment: The general-purpose haptic device has been studied and developed for many years, e.g., ‘Development of a Hybrid Haptic Device with a High Degree of Motion decoupling’. . Therefore, the above statement is bot accurate.
Comment 3:
Quoted: “This approach pioneeringly combines the fuzzy logic module for estimating the stiffness of grasped objects with the dynamic compliant primitives based self-adaptive regulation of impedance stiffness, in the context of teleoperated hand manipulation. This integration enables adaptive impedance control that closely mimics human muscle behavior, enhancing the system’s ability to handle objects of different stiffness safely and effectively.”
Comment:
- Fuzzy logic control or part of control is widely used in robotic control. Why is this claimed to be a pioneer work?
- Please clarify the concept of stiffness, as stiffness is a tensor rather than a scalar number. Why is the stiffness varying?
Comment 4:
The phrase ‘Compliant hand’ is very confusing, because in literature, researchers refer to compliant hand as the hand made to be deformable or soft. Please check literature, e.g., ‘Topology optimization of a fully compliant prosthetic finger: design and testing’, ‘Flexure hinge based fully compliant prosthetic finger’. Perhaps control of these soft or compliant hands will be future work.
Comment 5:
The novelty of the work in this paper needs more justified. In general, there does not seem to be with new idea or new method in the work.
Author Response
We appreciate the time and effort you dedicated to providing feedback on our manuscript and are grateful for the insightful comments and valuable improvements to our paper. We have incorporated most of the reviewers’ suggestions, which are in red within the revised manuscript. Please see below, for a point-by-point response to the reviewers’ comments and concerns.
Comments 1:
Quoted: “Shared control is widely applied in fields such as robotics [2], autonomous vehicles [3], teleoperation [4–6], and assistive technologies [7].”
Comment: Shared control is part of robotics. Therefore, the application of shared control should include rehabilitation, e.g., ‘Design and Rapid Construction of a Cost-Effective Virtual Haptic Device. ‘
Response 1:
Thank you for your valuable suggestion. We agree with this comment. Therefore, we modify the original sentence to “Shared control is widely applied in fields such as robotics [2], autonomous vehicles [3], teleoperation [4-6], and rehabilitation [7-9].”, which is in lines 29-30 of the revised manuscript with track changes. And ‘Design and Rapid Construction of a Cost-Effective Virtual Haptic Device. ‘ is added to the references.
Comments 2:
Quoted: “However, there are still some problems with current hand teleoperation technology, the first is that most hand teleoperation is without force feedback, some of them use data gloves without force feedback.”
Comment: The general-purpose haptic device has been studied and developed for many years, e.g., ‘Development of a Hybrid Haptic Device with a High Degree of Motion decoupling’. Therefore, the above statement is bot accurate.
Response 2:
Thank you for your valuable suggestion. We agree that the statement is not entirely accurate. Our intention is to convey that most hand teleoperation systems either lack force feedback or do not provide precise finger-to-finger force feedback, which results in a poor sense of immersion during hand teleoperation. For instance, the haptic device described in the article ‘Development of a Hybrid Haptic Device with a High Degree of Motion decoupling’ cannot achieve finger-to-finger mapping or provide finger-to-finger force feedback. We have revised this in lines 40-48 of the revised manuscript with track changes. Besides, ‘Development of a Hybrid Haptic Device with a High Degree of Motion decoupling’ is added to the references.
Comments 3:
Quoted: “This approach pioneeringly combines the fuzzy logic module for estimating the stiffness of grasped objects with the dynamic compliant primitives based self-adaptive regulation of impedance stiffness, in the context of teleoperated hand manipulation. This integration enables adaptive impedance control that closely mimics human muscle behavior, enhancing the system’s ability to handle objects of different stiffness safely and effectively.”
Comment:
- Fuzzy logic control or part of control is widely used in robotic control. Why is this claimed to be a pioneer work?
- Please clarify the concept of stiffness, as stiffness is a tensor rather than a scalar number. Why is the stiffness varying?
Response 3:
1. Thank you for pointing this out. Indeed, fuzzy logic control is widely utilized in the field of robotic control. In this article, we combine fuzzy logic control with dynamic compliant primitives, resulting in notable compliance improvements in the teleoperation of dexterous robotic hands. It is important to emphasize that the Fuzzy Logic-Dynamic Compliant Primitives controller represents a pioneering innovation, offering a novel approach that goes beyond the application of fuzzy logic control alone.
2. Thank you for pointing this out. In the context of this article, the term ‘stiffness’ is employed to denote two distinct concepts. In the first instance, stiffness refers to the measure of an object's resistance to deformation when being grasped. Generally, stiffness is represented as a tensor; however, for the purposes of simplification in this article, only the stiffness along one direction is considered. Thus, the stiffness of an object in this scenario is treated as a constant value.
The second usage pertains to the stiffness coefficient within the framework of impedance control, which determines the extent to which a robot resists external forces. A high stiffness coefficient indicates that the robot is highly sensitive to positional errors and will apply greater force to quickly correct these deviations. Conversely, a low stiffness coefficient signifies a higher tolerance for positional errors, leading the robot to more readily conform to external forces. To achieve human-like dexterity and compliance during grasping, variable impedance control is adopted in this article. Therefore, the stiffness here serves as a parameter in impedance control that needs to be dynamically adjusted.
In summary, while both usages of stiffness are related to the concept of resisting deformation, they pertain to different contexts: one describes the intrinsic property of the grasped object, and the other is a control parameter that facilitates adaptive interaction between the robot and its environment through variable impedance control. To distinguish more clearly, we replaced ‘stiffness’ with ‘stiffness coefficient’ of impedance control in the revised manuscript.
Comments 4:
The phrase ‘Compliant hand’ is very confusing, because in literature, researchers refer to compliant hand as the hand made to be deformable or soft. Please check literature, e.g., ‘Topology optimization of a fully compliant prosthetic finger: design and testing’, ‘Flexure hinge based fully compliant prosthetic finger’. Perhaps control of these soft or compliant hands will be future work.
Response 4:
Thank you for pointing this out. The phrase ‘compliant hand’ does not refer to the hand made to be deformable or soft in the article; rather, ‘compliant’ modifies the method, indicating that this teleoperation method proposed is compliant. Many hand teleoperation methods do not consider compliance, they are prone to causing damage to the objects being manipulated and the dexterous hands. Therefore, we innovatively proposed this aspect. We are sorry for any misunderstanding caused. Besides, we have changed the title into ‘A Novel Hand Teleoperation Method with Force and Vibrotactile Feedback Based on Dynamic Compliant Primitives Controller’ for simplicity. And we have changed ‘compliant hand teleoperation method’ into ‘compliant teleoperation method of hand’ in the revised manuscript text.
Comments 5:
The novelty of the work in this paper needs more justified. In general, there does not seem to be with new idea or new method in the work.
Response 5:
We sincerely appreciate your valuable feedback. Regarding the concern about the novelty of our work, we would like to clarify and emphasize the following innovative aspects that distinguish our approach from existing methods.
Traditional hand teleoperation systems typically employ position control method directly. However, this approach often lacks the compliance necessary for delicate or complex manipulation tasks, leading to challenges in achieving smooth and adaptive interactions. The Fuzzy Logic-Dynamic Compliant Primitives teleoperation method for dexterous hands that we propose is the first to enable adaptive adjustment of impedance control stiffness coefficients when grasping various objects. This method adjusts the stiffness coefficient of impedance control according to objects: it increases for harder objects and decreases for softer objects. Moreover, it achieves a human-like, gradual change in stiffness coefficients during the grasping process. This capability ensures adaptive compliance, effectively protecting both the grasped objects and the dexterous hand itself from potential damage.
Additionally, our control framework integrates finger-to-finger force feedback and vibrotactile feedback, significantly enhancing the operator's sense of immersion during teleoperation compared to other methods. By improving the success rate of teleoperation tasks and reducing operator fatigue, our approach represents a significant advancement in teleoperated manipulation systems. These innovations make it possible to perform complex tasks more intuitively and efficiently, paving the way for broader applications in remote and hazardous environments.
Round 2
Reviewer 1 Report
Comments and Suggestions for Authors
Thanks for the point-by-point responses. All comments have been responded to. The manuscript quality is enhanced based on the revised version. The paper can be accepted for publication.
Comments on the Quality of English LanguageThe term 'stiffness coefficient' is used more than 45 times throughout the paper. The authors should make an effort to improve the readability and writing quality of the manuscript.
Reviewer 2 Report
Comments and Suggestions for Authors
I am satisfied with the revision.